# EquAct: An SE(3)-Equivariant Multi-Task Transformer for 3D Robotic Manipulation

**Xupeng Zhu, Yu Qi**[*]**, Yizhe Zhu,**[*] **Robin Walters**[†]**, Robert Platt**[†]
Khoury College of Computer Sciences
Northeastern University
Boston, MA, 02115
`{zhu.xup, qi.yu2, zhu.yizhe, r.walters}@northesatern.edu,`
`rplatt@ccs.neu.edu`

## Abstract

Multi-task manipulation policy often builds on transformer's ability to jointly process language instructions and 3D observations in a shared embedding space. However, real-world tasks frequently require robots to generalize to novel 3D object poses. Policies based on shared embedding break geometric consistency and struggle in 3D generation. To address this issue, we propose EquAct, which is theoretically guaranteed to generalize to novel 3D scene transformations by leveraging SE(3) equivariance shared across both language, observations, and action. EquAct makes two key contributions: (1) an efficient SE(3)-equivariant point cloud-based U-net with spherical Fourier features for policy reasoning, and (2) SE(3)-invariant Feature-wise Linear Modulation (iFiLM) layers for language conditioning. Finally, EquAct demonstrates strong spatial generalization ability and achieves state-of-the-art across 18 RLBench tasks with both SE(3) and SE(2) scene perturbations, different amounts of training data, and on 4 physical tasks. Code is available in `https://github.com/ZXP-S-works/EquAct`.

## 1 Introduction

Recent breakthroughs in multi-task keyframe action policy learning (Shridhar et al., 2023; Goyal et al., 2024; Fang et al., 2025) have been driven by the success of transformer architectures (Vaswani et al., 2017), which excel at bridging different modalities by tokenizing language, 3D observations, and next-best keyframe actions into a shared embedding space. However, real-world robotic tasks often involve substantial SE(3) variation in object poses—for example, harvesting fruit from branches at diverse orientations, performing pipe work with fixtures mounted on walls or ceilings, and assembling components onto pegs at angles. While transformers excel at cross-modal integration, their tokenization process discards the underlying 3D geometric structure. Consequently, existing multi-task keyframe action methods struggle to generalize to novel 3D scene configurations and require large amounts of robot data to learn geometric priors from scratch.

This paper presents EquAct, a novel multi-task transformer that is theoretically guaranteed to generalize to novel 3D scene transformations. EquAct adapts actions SE(3)-equivariantly with 3D scene transformations and SE(3)-invariantly when language instructions remain unchanged. This adaptation is achieved by introducing a novel SE(3)-equivariant point transformer U-net with field networks for keyframe action evaluation, alongside novel SE(3)-invariant FiLM (iFiLM) layers to condition the policy on language in a semantically dependent yet geometrically invariant way.

EquAct is the first method to achieve continuous SE(3)-equivariance (covering both 3D rotation and translation) in a single unified model for multi-task policy. In contrast, previous SE(3)-equivariant methods (Simeonov et al., 2022; Ryu et al.; 2024; Huang et al., a; Zhu et al., 2025b) are single task, and multi-task methods (Zhu et al., 2025a; Gervet et al., 2023; Ke et al.) are only translationally equivariant. Moreover, to reflect that in realistic tasks, objects often have random 3D poses, rether

---

[*]Equal contribution.
[†]Equal advising.

than random 2D poses in RLBench, we proposes RLBench tasks with $SE(3)$ initialization. EquAct achieving state-of-the-art performance on RLBench $SE(2)$ and $SE(3)$ benchmarks, and on 4 physical experiments. Practically, our method leverages a spherical Fourier representation, and achieves computational efficiency for training and inference, matching the computation overhead of non-equivariant baselines.

To summarize, the contributions of this paper are as follows:

1. We propose a continuous $SE(3)$-equivariant multi-task policy with a novel equivariant U-net architecture, novel invariant FiLM layers, and novel equivariant field networks.

2. We mathematically prove the relevant equivariance and invariance properties.

3. We verify that EquAct outperform baselines across on RLBench with 18 tasks and 249 language goals with $SE(2)$ or $SE(3)$ initialization, and on 4 physical tasks.

## 2 RELATED WORKS

**Keyframe action and multi-task manipulation policy.** Keyframe action formulation was first introduced by (James & Davison, 2022), which approximates closed-loop manipulator trajectories using a sequence of discrete keyframes, thereby simplifying policy learning. Building on this idea, PerAct (Shridhar et al., 2023) proposes a transformer-based agent that learns a multi-task policy—executing different keyframe actions conditioned on natural language instructions. Later, the multi-task policy learning has diverged into two main directions to evaluate translational action. The first class consists of multi-view-based methods (Goyal et al., 2023; 2024; Wang et al., 2024b; Zhang et al., 2025; Fang et al., 2025), where the 3D scene is projected into three orthogonal image planes, followed by a ViT-like (Dosovitskiy et al., 2020) multi-view transformer that evaluates translational action values. While this approach is computationally efficient, reasoning in the image plane sacrifices geometric fidelity and requires clever strategies to project into $SE(3)$ (Xu et al., 2024) or $SO(3)$ (Klee et al., 2023; Park et al., 2022) space to achieve $SE(3)$-equivariance. The second class operates directly in 3D space (Gervet et al., 2023; Xian & Gkanatsios, 2023; Ke et al.; Garcia et al., 2025), typically using point-cloud-based transformers with densely sampled query points or diffusion models (Chi et al., 2023) to evaluate translational actions. These methods can achieve 3D translational equivariance through 3D CNNs or relative positional embeddings (Su et al., 2024), but are not 3D relationally equivariant. For rotational action prediction, existing approaches typically rely on discretized Euler angles or denoise $SO(3)$ rotations. While the former suffers from gimbal lock and discontinuity issues (Zhou et al., 2019), the latter incurs significant computational overhead due to iterative refinement. In contrast, EquAct achieves both translation and rotation equivariance. It also achieves fast inference by evaluating actions in one shot.

**Equivariant policy learning.** Previous works (Van der Pol et al., 2020; Wang et al., 2022b) have shown that geometric structures are inherent in reinforcement learning problems and that incorporating equivariant policy learning can lead to improved performance. Building on this insight, a series of methods (Zeng et al., 2018; Wang et al., 2021; Zhu et al., 2022; Huang et al., 2022; Wang et al., 2022a; Zhu et al., 2023; Liu et al., 2023; Wang et al., 2023; Nguyen et al., 2023; Huang et al., 2023a; Zhao et al., 2023; Jia et al., 2023; Huang et al., a; Kohler et al., 2024; Wang et al., 2024a; Tangri et al., 2024; Hu et al., 2025) have proposed $SE(2)$-equivariant policy learning for robotic tasks. More recently, (Simeonov et al., 2022; Ryu et al.; Huang et al., 2023b; Ryu et al., 2024; Hu et al.; Gao et al., 2024; Zhu et al., 2025a; Yang et al., 2024; Huang et al., b; Qi et al., 2025; Yang et al.; Tie et al., 2025; Zhu et al., 2025b) extended equivariance to the full $SE(3)$ group. However, all of these equivariant policy learning methods are limited to single-policy learning. In contrast, EquAct learns multi-task, language conditioned keyframe policies using a single unified model.

**Equivariant neural networks.** There are several approaches to achieving equivariance in learning-based robotic policies. A common method is data augmentation (Laskin et al., 2020), where both inputs and outputs are transformed according to the desired group symmetry during training. Another strategy is canonicalization (Zeng et al., 2018), which aligns inputs to a canonical frame prior to inference. An alternative is to leverage *equivariant neural networks*, which incorporate equivariance directly into the architecture through symmetry-preserving operations. Prior works (Wang et al., 2021; Zhu et al., 2022; Miller et al., 2020) have shown that such networks outperform data augmentation

and canonicalization by a significant margin. Equivariant neural networks are grounded in rigorous math from group theory, enabling them to preserve symmetry while maintaining high expressiveness. One class of such networks leverages *group convolutions* (Cohen & Welling, 2016; Weiler & Cesa, 2019; Cesa et al., 2022), which typically discretize a symmetry group and apply convolution over its elements. However, these approaches may suffer from discretization artifacts. Another class operates in the Fourier domain (Geiger & Smidt, 2022; Liao & Smidt, 2023; Passaro & Zitnick, 2023; Liao et al., 2024), which offers a more compact and continuous representation of the group. Building on this Fourier-based framework, our method achieved natural language conditioning and fast SE(3) action inference.

**Equivariant natural language processing.** Incorporating natural language into equivariant models has recently gained attention (Li et al., 2025; Roche et al., 2024; Jia et al., 2024). Li et al. (2025) and Roche et al. (2024) combine equivariant graph neural networks with invariant language embeddings and evaluate their effectiveness at scale. However, they are either limited to molecule generation or $SE(2)$ equivariance. In contrast, our work is the first to explicitly identify the SE(3) invariance of natural language instructions in the context of robotic policies, and introduce simple yet effective invariant FiLM (iFiLM) layers to enforce this invariance within an SE(3)-equivariant policy network.

## 3 BACKGROUND

**Equivariant policy learning.** A function $f$ is equivariant with respect to a group $G$ if the group action $g \in G$ commutes with the function, i.e., $f(g \cdot x) = g \cdot f(x)$. In this paper, we focus on the special Euclidean group $SE(3) = SO(3) \ltimes \mathbb{T}(3)$, which represents 3D rigid-body transformations composed of 3D rotations $SO(3)$ and translations $\mathbb{T}(3)$. An equivariant robotic policy (Wang et al., 2022c;b) satisfies the property:

$$\pi(g \cdot o) = g \cdot \pi(o), \tag{1}$$

meaning the action transforms as the observation transforms. For example, an SE(2)-equivariant planar grasping policy (Zhu et al., 2022) predicts a grasp pose from an input image; if the image is rotated, the predicted grasp pose rotates accordingly. There are several strategies to enforce equivariance in neural network-based policies. One common approach is data augmentation (Laskin et al., 2020; Wang et al., 2022c), where both observations and corresponding actions are transformed according to Equation 1 during training. Another method is canonicalization, which transforms the input into a standard reference frame aligned with the action space (Zeng et al., 2018). More recently, robot policies that leverage equivariant neural networks (Wang et al., 2022c; Zhu et al., 2022; Huang et al., 2023b; Weiler & Cesa, 2019; Deng et al., 2021; Zhu et al., 2025b) have been shown to outperform these alternatives by embedding equivariance directly into the network architecture, but none of them studies multi-task policy learning.

**Spherical harmonics.** SE(3)-equivariant models rely on feature representations based on spherical functions and spherical harmonics. A spherical function $f_s \colon S^2 \to \mathbb{R}$ maps a point on the sphere $u \in S^2$ to a real value $y$. An alternative representation of $f_s$ is its Fourier form, where the function is decomposed into spherical harmonic coefficients $c_l^m$ via the spherical Fourier transform $\mathcal{F} \colon f_s \mapsto \hat{f}_s$, such that $\hat{f}_s = \{c_l^m\}$. Each coefficient $c_l^m$ denotes the weight of the corresponding spherical harmonic $Y_l^m \colon S^2 \to \mathbb{R}$, which forms an orthonormal basis for the function space $L^2(S^2, \mathbb{R})$. These basis functions are indexed by type (or degree) $l \in \mathbb{Z}_{\geq 0}$ and order $m \in \mathbb{Z}$ such that $-l \leq m \leq l$. The inverse spherical Fourier transform reconstructs the spatial function as $\mathcal{F}^{-1}(f_s)(u) = \sum_{l=0}^{\infty} \sum_{m=-l}^{l} c_l^m Y_l^m(u)$. In practice, truncated spherical coefficients $l \leqslant L_{max}$ are used because they provide a good approximation of the spherical function (Liao & Smidt, 2023). Spherical functions are steerable under $SO(3)$, making them well-suited for $SO(3)$-equivariant neural networks (Thomas et al., 2018; Liao & Smidt, 2023; Fuchs et al., 2020; Passaro & Zitnick, 2023; Liao et al., 2024). Specifically, rotating the input function by $g \in SO(3)$, i.e., $f_s'(u) = g \cdot f_s(u) = f_s(g^{-1}u)$, corresponds to rotating its Fourier coefficients via the Wigner D-matrices D: $c_l^{n'} = \sum_m \mathrm{D}_{mn}^l(g) c_l^m$, where $c_l^{n'}$ are the coefficients of the rotated function $f_s'$. For example, a type-0 feature is a scalar, and its Wigner D-matrix is identity; a type-1 feature is a 3D vector, and its Wigner D-matrix is a 3D rotation matrix.

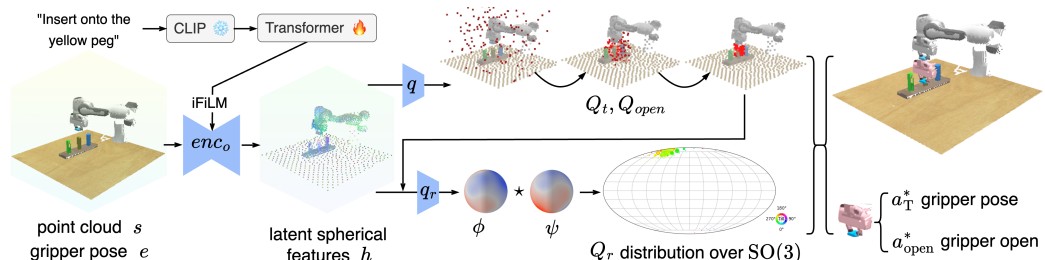

Figure 1: **Overview of EquAct.** EquAct first encodes the observation $o = \{s, e\}$ into latent spherical features $h$ using a SE(3)-equivariant U-Net, $enc_o$, while conditioning the natural language instruction $n$ through invariant iFiLM layers. Based on the encoded features $h$, EquAct then samples and refines translational query actions and gripper open actions using an equivariant field network, resulting in action value functions $Q_t$ and $Q_{open}$. Finally, a rotational field network aggregates spherical features from $h$ centered at the predicted translation $a_t^*$ to obtain a latent feature $\phi$, which is subsequently convolved with a learned filter $\psi$ to produce the rotational action value function $Q_r$.

**Spherical CNN.** Spherical Convolutional Neural Networks (Cohen et al., 2018) lift a spherical function $f_s$ to an SO(3) function $f_{\text{SO}(3)} \colon \text{SO}(3) \to \mathbb{R}$ by convolving it with a learnable spherical filter $\psi$ as such $(f_s \star \psi)[g] = \int_{S^2} f_s(u)\psi(g^{-1} \cdot u)\,du, g \in \text{SO}(3)$. This spatial convolution is equivalent to an outer product in the Fourier domain: $\widehat{f_s \star \psi} = \hat{f}_s \cdot \hat{\psi}$, which is more efficient than performing the convolution directly in the spatial domain (Cohen et al., 2018; Klee et al., 2023).

**Multi-task keyframe policy formulation.** Following PerAct (Shridhar et al., 2023), we formulate multi-task keyframe manipulation policy learning as a mapping from an observation $o$ and a natural language instruction $n$ to the next best keyframe action of the gripper $a$, denoted as $\pi(o, n) = a$. Then a motion planner generates a trajectory to reach this keyframe action. This formulation decomposes robot trajectories into a sequence of keyframe poses, thus simplifies learning. For example, PerAct learns multi-task keyframe policy using 53 demonstrations, but (Team et al., 2025) relied on $64, 262$ demonstrations to learn trajectory policy. The observation $o = \{s, e\}$ consists of the scene information $s$, and the end-effector state $e$. The scene $s$ is represented as a colored point cloud of 2500 points generated from $256 \times 256$ RGBD images captured by calibrated front, left, right, and in-hand cameras. The end-effector state $e$ or action $a$ are expressed as $\xi = \{\xi_T, \xi_{open}\}, \xi = e$ or $\xi = a$, where $\xi_T \in \text{SE}(3)$ denotes the gripper pose and $\xi_{open} \in \{0, 1\}$ indicates whether the gripper is closed or open. The instruction $n$ is represented as a natural language string. The policy is trained via imitation learning: we first collect expert demonstrations $D$ consisting of observations, natural language goals, and expert actions, and then train the policy to predict the expert actions. At evaluation time, we test the policy on the training tasks but with novel object poses. For more details, please see Appendix C.

## 4 METHOD

EquAct is a multi-task keyframe action policy represented as an implicit function $Q_a(o, n, a) \in \mathbb{R}$ that estimates the action value given an observation $o$, a language goal $n$, and a query action $a$. The inference procedure is illustrated in Figure 1 and has the following steps. **1) The** SE(3)-**Equivariant Point Transformer U-Net** encodes the observation $o$ that includes a point cloud $s$ and the gripper pose $e$ into a set of latent spherical features $h$ at each point in the cloud $h = enc_o(o)$. **2) Invariant Feature-wise Linear Modulation layers** fuse the language embedding $k$, which is treated as type-0 features, into the U-Net. Here, $k$ is the encoding of the natural language instruction $n$, by using a frozen CLIP (Radford et al., 2021) tokenlizer and a Transformer (Vaswani et al., 2017) encoder. **3) The Equivariant Field Network** takes the latent point

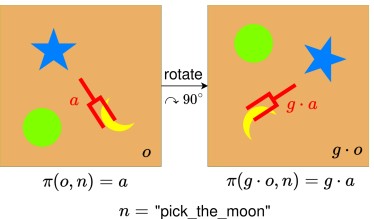

Figure 2: **The equivariance and invariance of the multi-task keyframe policy.** Under the equivariance assumption, when the observation is transformed to $g \cdot o$, the predicted action transforms accordingly to $g \cdot a$. Under the invariance assumption, given a fixed natural language instruction $n$, the action transformation depends solely on the transformation applied to the observation.

cloud $h$ and sampled query actions $a = \{a_t, a_{open}, a_r\}$ as input and predicts values for each action $q(a, h) \in \mathbb{R}$. The final output actions $a_t^*, a_r^*$ and $a_{\text{open}}^*$ are chosen as those with the highest action values.

During training, EquAct minimizes the following loss:

$$\mathcal{L} = \mathbb{E}_{D,A}\Big[\mathcal{H}(Q_t(a_t, o, n), \bar{a}_t) + \mathcal{H}(Q_r(a_r, \bar{a}_t, o, n), \bar{a}_r) + \mathcal{H}(Q_{open}(a_{open}, \bar{a}_t, o, n), \bar{a}_{open})\Big],$$

where $(o, n, \bar{a}) \sim D$ are expert demonstrations consisting of observations $o$, natural language instructions $n$, and expert actions $\bar{a}$, and $a \sim A$ denotes uniformly sampled query actions from the action space. Specifically, $a_t$ consists of $449$ uniformly sampled translational actions, and $a_r$ consists of 36,864 rotational actions sampled using the HEALPix (Gorski et al., 2005; Klee et al., 2023) grid. $\mathcal{H}$ denotes cross-entropy loss. Intuitively, this loss treat policy learning as a classification problem in which the goal is the policy to correctly choose the expert action from among all available actions. During training, we also augment the dataset with respect to equation 1 by randomly rotating the point cloud and the action simultaneously with $[\pm 5°, \pm 5°, \pm 45°]$ rotation along $[x, y, z]$ axis.

## 4.1 EQUIVARIANCE ASSUMPTIONS IN MULTI-TASK MANIPULATION POLICY LEARNING

EquAct assumes that the keyframe action policy is equivariant with respect to the observation. That is, when the observation undergoes a transformation, the predicted action should transform accordingly (Wang et al., 2021; Zhu et al., 2022; Huang et al., 2022; Ryu et al.). Additionally, we identify and assume that the action is invariant to the natural language instruction—meaning that for a fixed instruction, the action should transform solely based on $\mathrm{SE}(3)$ transformation of the observation. Formally, this behavior is expressed as:

$$\pi(g \cdot o, n) = g \cdot a, \quad g \in \mathrm{SE}(3), \tag{2}$$

where the group action $g$ operates on both the observation $o$ and the predicted action $a$ by applying rigid-body transformations to the point cloud $s$ or the gripper poses $e_\mathrm{T}$ and $a_\mathrm{T}$, see Figure 2 for an illustration.

Methodologically, EquAct achieves equivariance between the observation and action by employing a novel $\mathrm{SE}(3)$-equivariant Point Transformer U-Net (Section 4.2) and $\mathrm{SE}(3)$-equivariant field networks (Section 4.4). In parallel, it enforces invariance with respect to natural language instructions via the proposed $\mathrm{SE}(3)$-invariant layer-wise modulation (iFiLM) layers (Section 4.3).

**Proposition 4.1.** *EquAct is* $\mathrm{SE}(3)$*-equivariant in observation-action mapping and* $\mathrm{SE}(3)$*-invariant to nature language instruction, as described in Equation 2.*

This is proved by induction; see Appendix B.1.

## 4.2 EQUIVARIANT POINT TRANSFORMER U-NET (EPTU)

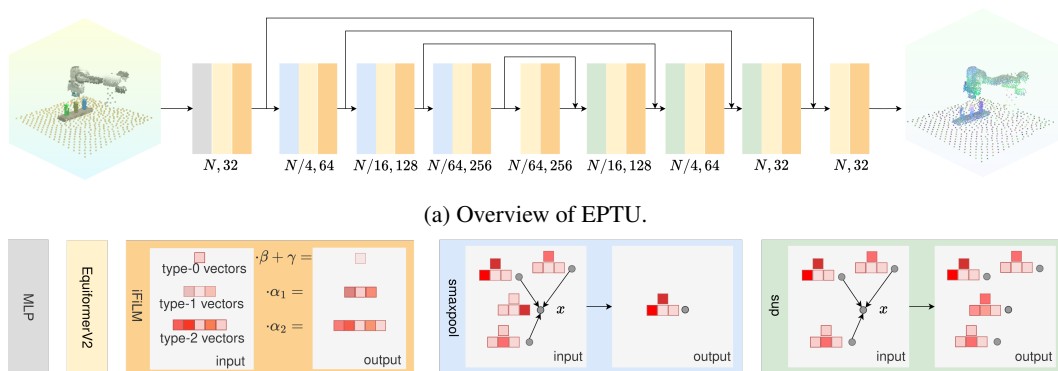

(a) Overview of EPTU.

(b) Detailed structure design for each module. Red color indicates the magnitude of the feature.

Figure 3: $\mathrm{SE}(3)$-**Equivariant Point Transformer U-net (EPTU).**

The $SE(3)$-equivariant Point Transformer U-Net (EPTU, Figure 3) encodes a point cloud $s$ into equivariant latent features by propagating both local and global information across points. Compared to non-equivariant counterparts such as Point Transformer (Zhao et al., 2021) and U-Net (Ronneberger et al., 2015), EPTU achieves continuous $SE(3)$-equivariance by leveraging spherical Fourier features in its hidden layers. EPTU further improves the computational efficiency of EquiformerV2 (Liao et al., 2024) by adopting a U-net-style architecture (Ronneberger et al., 2015), which incorporates novel *spherical Fourier maxpooling* layers to compress point cloud features and *spherical Fourier upsampling* layers to reconstruct features back to the original resolution. These pooling and upsampling layers are interleaved with standard EquiformerV2 (Liao et al., 2024) graph attention blocks, which first construct a $k$-nearest-neighbor graph for each point and then apply equivariant attention-based message passing. EPTU also incorporates skip connections (Ronneberger et al., 2015) between the downsampling and upsampling stages. Compared to prior equivariant point U-Net (Ryu et al., 2024; Hu et al.), the proposed U-Net is straightforward to implement, by eliminating the need for caching graphs, i.e., the sup block in Figure 3 (b) does not need the maxpool graph in the smaxpool model.

**Spherical Fourier maxpooling.** Analogous to the maxpooling operation in convolutional neural networks (LeCun et al., 1998), the *spherical Fourier maxpooling* layer (Figure 3 (b) middle) reduces the resolution of the feature map in the spherical Fourier domain. Specifically, for a point $x$, the layer aggregates features from its $k$-nearest neighborhood $\{c_{l,p} \mid p \in knn(x)\}$ and selects the spherical Fourier coefficient with the largest magnitude at each degree $l$:

$$c'_{l,x} = \mathrm{smaxpool}\{c_{l,p} \mid p \in knn(x)\} = c_{l,p^*}, \quad p^* = \arg\max_{p \in knn(x)} \|c_{l,p}\|_2^2, \tag{3}$$

where the $2l+1$-dimension vector $c_{l,p} = [c_{l,p}^{-l}, c_{l,p}^{-l+1}, \ldots, c_{l,p}^{l}]$ denotes the type-$l$ spherical Fourier coefficient at point $p$.

**Proposition 4.2.** *The spherical Fourier maxpooling operation defined in Equation 3 is* $SE(3)$-*equivariant. That is, for any $r \in SO(3)$ and $t \in \mathbb{T}(3)$:*

$$\mathrm{D}(r) \cdot c'_{l,\, t+x} = smaxpool\{\mathrm{D}(r) \cdot c_{l,p} | p \in t + knn(x)\}. \tag{4}$$

This is proved by the orthogonal property of Wigner D-matrices, see Appendix B.2 for a proof.

**Spherical Fourier upsampling.** Interpolation is commonly used for upsampling feature maps (Zhao et al., 2021; Ronneberger et al., 2015). To extend this operation to the spherical Fourier domain, we propose a novel *spherical Fourier upsampling* method (Figure 3 (b) right). Specifically, for each type-$l$ component, we perform a coefficient-wise interpolation over the $k$-nearest neighbors of a query coordinate $x$:

$$c'_{l,x} = \sup\{c_{l,p}, x | p \in \mathrm{knn}(x)\} = \mathrm{softmax}_{p \in \mathrm{knn}} \left( \frac{1}{\|x - p\|} \right) c_{l,p}, \tag{5}$$

**Proposition 4.3.** *The spherical Fourier upsampling operation defined in Equation 5 is* $SE(3)$-*equivariant. Specifically, for any $r \in SO(3)$ and $t \in \mathbb{T}(3)$:*

$$\mathrm{D}(r) \cdot c'_{l,\, t+x} = sup\{\mathrm{D}(r) \cdot c_{l,\, p},\, t + x | p \in t + knn(x)\}. \tag{6}$$

See Appendix B.3 for a proof. The proof is based on Schur's lemma (Schur, 1905) and that the linear $SO(3)$ action on the Fourier coefficients.

## 4.3 INVARIANT FEATURE-WISE LINEAR MODULATION LAYERS (iFiLM)

We propose invariant Feature-wise Linear Modulation (iFiLM) layers (Figure 3 (b) left) to enforce the geometric invariance of natural language conditioning in the policy, as defined in Equation 2. Unlike standard FiLM layers (Perez et al., 2018), which do not guarantee equivariance or invariance, the iFiLM layer is provably $SE(3)$ invariant with respect to the conditioning input $k$. Specifically, the iFiLM layer takes as input a spherical Fourier feature $c$ and a type-0 (invariant) condition feature $k$, and outputs a semantically modulated feature $c'$:

$$c' = \mathrm{iFiLM}(c, k), \quad \alpha_l, \beta, \gamma = \mathrm{MLP}(k), \tag{7}$$

$$c'_l = \alpha_l c_l, \quad \text{for } l > 0, \tag{8}$$

$$c'_0 = \beta c_0 + \gamma, \quad \text{for } l = 0, \tag{9}$$

where iFiLM first uses a multi-layer perceptron to project the condition $k$ into type-0 modulation scales $\alpha, \beta$ and bias $\gamma$. Then, iFiLM scales the type-$l$ input feature $c_l$ by $\alpha_l$ for all $l > 0$, and applies an affine transformation to the type-0 features using $\beta$ and $\gamma$.

**Proposition 4.4.** *The invariant feature-wise linear modulation (iFiLM) layer is* SO(3)-*invariant with respect to the condition input* $k$, *and* SO(3)-*equivariant with respect to the input feature* $c$. *Specifically, for any rotation* $r \in$ SO(3):

$$\mathrm{D}(r) \cdot c' = \mathrm{iFiLM}(\mathrm{D}(r) \cdot c, \, k). \tag{10}$$

See Appendix B.4 for a proof. The proof utilizes Shur's lemma (Schur, 1905).

## 4.4 EQUIVARIANT FIELD NETWORK

EquAct evaluates actions over the entire pose action space $A_\mathrm{T} \subset$ SE(3), rather than actions anchored at each point in the point cloud (Hu et al.). To achieve this, we introduce equivariant field networks $q$ that propagate features from the latent point cloud representation $h$ to any query point $a_\mathrm{T} \in A_\mathrm{T}$, where the action is decomposed into translational and rotational components, $a_\mathrm{T} = a_t \rtimes a_r$.

For translational action value evaluation, given the query translational action $a_t$ and the latent point cloud $h$, the field network $q_t$ builds a graph with $h$ as the source and $a_t$ as the destination, then performs graph attention to aggregate spherical Fourier features from $h$ to $a_t$. The graph connects the query point to the $k$-nearest neighbor in $h$. The graph attention is implemented by one EquiformerV2 attention block (Liao et al., 2024). The graph building and attention operation is similar to (Gervet et al., 2023; Ryu et al., 2024; Chatzipantazis et al., 2023), except that the output, i.e., the translation action value is invariant to rotation (Wang et al., 2021; Zhu et al., 2022): $q_t(a_t, h) = q_t(a_t, g \cdot h), g \in$ SO(3). Therefore, the field network only takes the type-0 feature from aggregated features. We evaluate the translational action in a coarse-to-fine fashion, where the initial resolution of action is coarse, and the subsequent sampling refines the action. The gripper open action $q_{open}(a_{open}, a_t, h)$ is evaluated in the same way, except that $q_{open}$ outputs two channels of type-0 features, corresponding to the open/close action values.

For rotational action value evaluation, given the query trans-rotal action $a_t, a_r$ and the latent point cloud $h$, the field network $q_r$ first aggregates features in the same way as the translational network to obtain the spherical Fourier features $\hat{\phi}$ at $a_t$. Then the action value for $a_r$ is calculated by a spherical CNN (Cohen et al., 2018) with a learnable filter $\hat{\psi}$: $q_r(a_r, a_t, h) = (\phi \star \psi)[a_r] = \mathcal{F}^{-1}(\hat{\phi} \cdot \hat{\psi})[a_r]$. Notice that our field network $q_r$ performs spherical convolution at a 3D location $a_t$ and is SE(3) equivariant, which differs from the previous SO(3) equivariant spherical convolution (Cohen et al., 2018; Klee et al., 2023; Howell et al., 2023) that operates in images.

# 5 EXPERIMENTS

## 5.1 SIMULATION EXPERIMENTS

**Task setups.** We benchmark multi-task algorithms on 18 RLBench (Shridhar et al., 2023; James et al., 2020) tasks. The benchmark uses a Franka Panda robot equipped with a parallel gripper. Observations are captured from four RGB-D cameras positioned at the front, left shoulder, right shoulder, and wrist, with resolutions of either $128^2$ or $256^2$ pixels. Each task includes several variations specified by natural language instructions. For example, in the "open_drawer" task, "open_the_top_drawer" and "open_the_middle_drawer" are two distinct variations. Across all tasks, there are between 2 and 60 variations per task, resulting in a total of 249 variations.

**Evaluation metric.** Performance is measured by a binary reward, where $0\%$ and $100\%$ correspond to failure and successful completion of the task according to the natural language instruction, respectively. We report the task success rate over 25 evaluation episodes per task, with a maximum of 25 steps per episode. During evaluation, the objects and language goals remain the same as in the training set, but the object poses are novel.

**Baselines.** We benchmark our method with two strong baselines. **SAM2ACT** (Fang et al., 2025) is the current state-of-the-art baseline on 18 RLBench, which leverages pretrained image tokenizer from

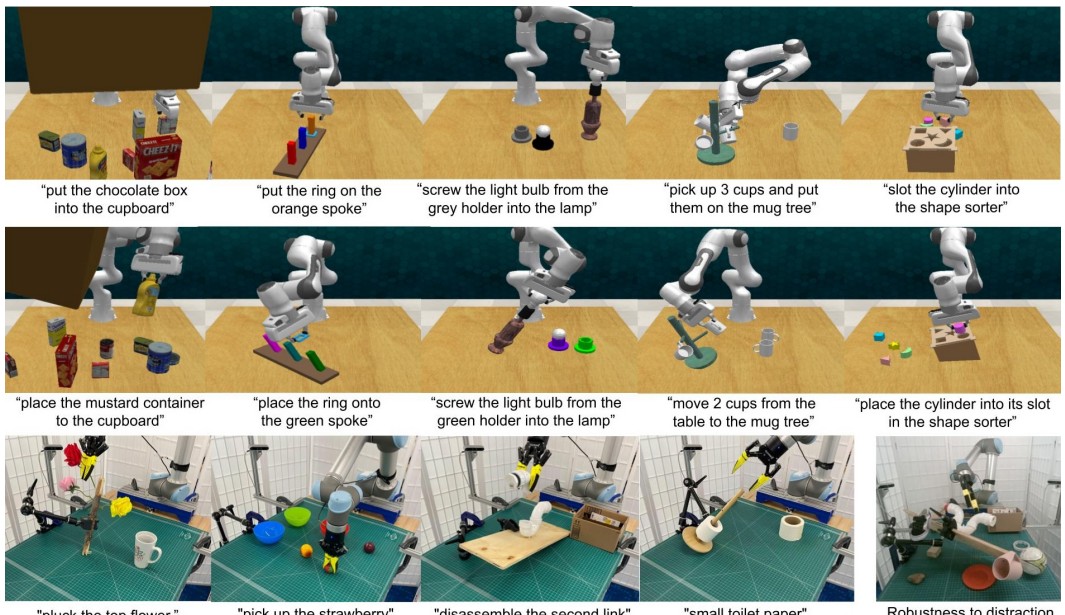

Figure 4: **Experiments setups.** First row: 18 standard RLBench tasks (Shridhar et al., 2023). Second row: 18 RLBench tasks with SE(3) random initialization. Third row: 3 SE(3) and 1 SE(2) physical experiments and a robustness test. A language instruction specifies each variant of the task.

Table 1: **Multi-task success rate (%) on** 18 **RLBench tasks with** 249 **instructions.** On average, EquAct outperforms all the baselines on all 3 settings. Furthermore, the second column shows that EquAct's training and inference time, GPU memory matches baselines. **2D/100** and **2D/10** denote 100 and 10 training demonstrations per task with object poses randomly initialized in SE(2). **3D/10** denotes task with object poses randomly initialized in SE(3) and 10 demonstrations per task.

| | avg. success rate ↑ | | | open drawer | | | slide block | | | sweep dust. | | | meat off grill | | |
|---|---|---|---|---|---|---|---|---|---|---|---|---|---|---|---|
| Method | 3D/10 | 2D/10 | 2D/100 | 3D/10 | 2D/10 | 2D/100 | 3D/10 | 2D/10 | 2D/100 | 3D/10 | 2D/10 | 2D/100 | 3D/10 | 2D/10 | 2D/100 |
| EquAct | **53.3** | **60.1** | **89.4** | **55** | 74 | 78 | **48** | 56 | **100** | 59 | 61 | 83 | **96** | **100** | **100** |
| SAM2ACT | 37.0 | 52.2 | 86.8 | 25 | 76 | 83 | 40 | 32 | 86 | 72 | 76 | **99** | 80 | 72 | 98 |
| 3DDA | 37.9 | 50.3 | 81.3 | 30 | **87** | **90** | 43 | **72** | 98 | **95** | **83** | 84 | **96** | 78 | 97 |

| | Train. | Infer. | Mem. | screw bulb | | | put in safe | | | place wine | | | put in cupboard | | |
|---|---|---|---|---|---|---|---|---|---|---|---|---|---|---|---|
| Method | t (h) ↓ | t (s) ↓ | (GB) ↓ | 3D/10 | 2D/10 | 2D/100 | 3D/10 | 2D/10 | 2D/100 | 3D/10 | 2D/10 | 2D/100 | 3D/10 | 2D/10 | 2D/100 |
| EquAct | 240 | 0.7 | 21 | **36** | 53 | 68 | **76** | **91** | **100** | 14 | **95** | 95 | 36 | 22 | **89** |
| SAM2ACT | 225 | 0.1 | 21 | 4 | **64** | **89** | 68 | 48 | 98 | 40 | 68 | 93 | 0 | 8 | 75 |
| 3DDA | 253 | 3.7 | 20 | 5 | 37 | 82 | 62 | 70 | 98 | **73** | 82 | 94 | 9 | **28** | 86 |

| | close jar | | | drag stick | | | stack blocks | | | stack cups | | | place cups | | |
|---|---|---|---|---|---|---|---|---|---|---|---|---|---|---|---|
| Method | 3D/10 | 2D/10 | 2D/100 | 3D/10 | 2D/10 | 2D/100 | 3D/10 | 2D/10 | 2D/100 | 3D/10 | 2D/10 | 2D/100 | 3D/10 | 2D/10 | 2D/100 |
| EquAct | **33** | 52 | 91 | **85** | 90 | 95 | **26** | **35** | **90** | 59 | 18 | 68 | **21** | **62** | **76** |
| SAM2ACT | 8 | **68** | **99** | 44 | **100** | 99 | 16 | 20 | 76 | 0 | 12 | **78** | 0 | 4 | 47 |
| 3DDA | 24 | 52 | 96 | 60 | 35 | **100** | 16 | 10 | 68 | 9 | 18 | 47 | 0 | 10 | 24 |

| | turn tap | | | put in drawer | | | sort shape | | | push buttons | | | insert peg | | |
|---|---|---|---|---|---|---|---|---|---|---|---|---|---|---|---|
| Method | 3D/10 | 2D/10 | 2D/100 | 3D/10 | 2D/10 | 2D/100 | 3D/10 | 2D/10 | 2D/100 | 3D/10 | 2D/10 | 2D/100 | 3D/10 | 2D/10 | 2D/100 |
| EquAct | **67** | 56 | **100** | 64 | 84 | **100** | **36** | 33 | **86** | **89** | 85 | **100** | **60** | 14 | **90** |
| SAM2ACT | 36 | **92** | 96 | 64 | **100** | 99 | 24 | 16 | 64 | 40 | 56 | **100** | 4 | **28** | 84 |
| 3DDA | 48 | 74 | 99 | 14 | 92 | 96 | 14 | **33** | 44 | 79 | **95** | 98 | 5 | 14 | 66 |

SAM2 (Ravi et al., 2024) and projects point cloud into image planes (Goyal et al., 2024). **3DDA** stands for 3D diffuser actor (Ke et al.), which takes point cloud as input and leverages diffusion policy to capture multi-modality in the demonstrations. All the baselines are trained and evaluated on a single RTX 4090 GPU with 24 GB memory. We report hyperparameters in Appendix G.

**Experiment settings.** We benchmark baselines on three experiment settings with increasing difficulty. In the 100 setting (Shridhar et al., 2023), the model is trained with 100 demonstrations per task, then tested with randomly SE(2) initialized objects. In the 10 setting, the model is trained with 10 demonstrations per task and tested in the same way as 100. In the 10 SE(3) setting, the training set contains 10 demo per task and both the training and testing scenes have randomly SE(3) initialized objects.

**Results.** Table 1 shows that on average, EquAct outperforms all the baselines on the 100 setting by 2.6%, the 10 setting by 6.2%, and the 10 SE(3) setting by 15.4%. Furthermore, the more difficult the setting is, the more EquAct outperforms the baselines, demonstrating strong sample efficiency and 3D generalization. EquAct also excels at tasks requiring precisions, e.g., "place_cups" and "sort_shape", where other baselines struggle. This indicates that the equivariance is crucial for a policy adapting precisely to objects pose. Lastly, EquAct underperforms baselines in the tasks in which the object's pose is fixed, e.g., "sweep_to_dustpan". Besides success rate, the sceond row in Table 1 shows that EquAct matches the training/inference time and GPU memory consumption of other baselines.

## 5.2 REAL-WORLD EXPERIMENTS

We benchmark the performance of EquAct and baseline on 4 physical multi-task with 11 variations and 135 demonstrations in total: "disassemble_pipe", "pluck_flower", "pick_fruit", "install_toilet_roll". The pose of objects in all tasks, except "pick_fruit", undergo random SE(3) transformations within the manipulator's workspace. Details of the experiment setting are given in Appendix F.

Table 2: **Real-world experiments.**

| Var × Demo | avg. SR ↑ | disass. pipe $3 \times 10$ | pluck flower $3 \times 15$ | pick fruit $3 \times 10$ | install toilet roll $2 \times 15$ |
|---|---|---|---|---|---|
| Ours | 65.0 | 90 | 70 | 50 | 50 |
| 3DDA | 12.5 | 0 | 20 | 30 | 0 |

We evaluate 10 episodes for each task and report the binary success rate. Notice that physical settings are more challenge than simulation, due to noisy demonstrations and noisy observations. We baseline with the best model 3DDA (Ke et al.) in the 10-SE(3) setting in Table 1, and show quantitative results in Table 2. EquAct effectively learns physical SE(3) multi-task keyframe policy from limited demonstrations, achieving 65% average success rate. In comparison, 3DDA struggles in these experiments, often skipping keyframe actions and resulting in failure.

## 5.3 ABLATION STUDY

We perform the ablations on the 10 demo setting: **Ours:** the full EquAct model. **aug. → no aug.** removes data augmentation by training with the raw demonstration data. **iFiLM → FiLM:** ablates iFiLM layers by replacing them with standard FiLM layers (Perez et al., 2018). $l = 3 \to 2$: reduces the spherical feature resolution in EquAct (reducing the spherical harmonic degree from 3 to 2). **EPTU → VN:** replaces the equivariant Point Transformer U-Net with a VN-DGCNN (Qi et al., 2025; Deng et al., 2021) network, which is also SE(3) equivariant. **equ. → no equ.:** breaks the equivariance by replacing one equivariant layer in $q_t$ and $q_r$ with a Roformer transformer layer (Su et al., 2024; Gervet et al., 2023).

Table 3: **Ablation study.**

| | avg. SR ↑ | place wine | place cups | reach drag | insert peg |
|---|---|---|---|---|---|
| Ours | 52.8 | 45 | 62 | 90 | 14 |
| aug. → no aug. | 50.5 | 36 | 71 | 85 | 10 |
| iFiLM → FiLM | 50.3 | 68 | 24 | 90 | 19 |
| $l = 3 \to 2$ | 45.5 | 64 | 28 | 80 | 10 |
| EPTU→VN | 22.0 | 64 | 5 | 5 | 14 |
| equ. → no equ. | 12.3 | 14 | 0 | 35 | 0 |

Table 3 reports the multi-task success rates across 4 RLBench tasks. Even though only a single equivariant layer is replaced, **equ. → no equ.** results in the largest performance drop, underscoring the critical role of geometric structure in EquAct. The **EPTU → VN** ablation shows that although VN-DGCNN is SE(3)-equivariant, it underperforms EquAct by more than 30%. A likely reason is that VN-DGCNN operates only on vector features (type-1 irreps), whereas EquAct leverages higher-order features (up to type-3). Similarly, the $l = 3 \to 2$ ablation highlights the importance of high-resolution spherical Fourier coefficients for accurate action reasoning. Additionally, replacing **iFiLM** with **FiLM** causes a notable drop on precision tasks (e.g., "place_cups"), confirming iFiLM's precision advantage, though **FiLM** can overfit on the tasks where actions are nearly constant. Finally,

**aug.** → **no aug.** indicates using data augmentation can further improve performance, we hypothesize that data augmentation reduces numerical error in the equivariant neural networks.

## 5.4 ROBUSTNESS TEST AND EMPIRICAL EQUIVARIANCE ERROR

**Robustness to occlusion.** To explore the degree to which our equivariance mitigates the effects of extrinsic corruptions such as partial observability, we performed an ablation study on 4 RLBench tasks given 10 demonstrations per-task (see Table 4). In the occluded setup, all models are trained and tested using only the front and in-hand cameras instead of all 4 cameras, resulting

Table 4: Robustness to occluded point cloud.

| Method | PCD | Avg. SR | Place Wine | Place Cups | Reach Drag | Insert Peg |
|--------|-----|---------|------------|------------|------------|------------|
| EquAct | Full | 52.8 | 45 | 62 | 90 | 14 |
| EquAct | Occluded | 47.0 | 50 | 43 | 96 | 0 |
| 3DDA | Occluded | 15.8 | 55 | 5 | 3 | 0 |

in significant occlusion in the point cloud. We found that EquAct's performance decreased by only $5.8\%$, suggesting that the model works well under this type of uncertainty. In contrast, 3DDA (Ke et al.) struggles when the observation is occluded.

**Robustness to distraction.** To test the robustness towards unseen distraction objects, we additionally evaluated the EquAct model in Section 5.2 on the "disassemble_pipe" task with 3 variations . In Table 5, we randomly placed 10 additional distraction objects in the scene, including (a toy car, a small soccer ball, a mug, a tape, a plate, etc.), and the performance of EquAct dropped from $90\%$ to $70\%$ success rate, demonstrating strong robustness in the cluttered scenes.

Table 5: Robustness to distracting objects.

| # Dist. Obj | Avg. SR | Disas. the 1st Link | Disas. the 2nd Link | Disas. All Links |
|-------------|---------|----------------------|----------------------|------------------|
| 0 | 90% | 3/3 | 3/3 | 3/4 |
| 10 | 70% | 3/3 | 2/3 | 2/4 |

**Equivariance error.** In addition to providing theoretical proofs of EquAct's equivariance in Section 4, we measure its empirical equivariance error in Appendix E. EquAct achieves lower $SE(3)$ equivariance error than 3DDA (Ke et al.). Together, the equivariance error, the equivariance proofs, and the consistent outperformance reported in Table 1 validate that equivariance is crucial for spatial generalization.

## 6 CONCLUSION AND LIMITATIONS

This paper proposes EquAct to leverage $SE(3)$ equivariance in the multi-task keyframe policy and invariance in the language instruction. Specifically we use a novel equivariant point transformer U-net (EPTU) to encode the observation and use equivariant field networks to evaluate action candidates. Then we propose invariant FiLM layers to modulate the policy with natural language instructions. In the end, EquAct outperforms SOTA baselines by $2.6\%$ and $6.2\%$ when trained with 100 or 10 demos in $SE(2)$ setting, and by $15.4\%$ when trained with 10 demos in $SE(3)$ setting. Physical experiments validated that EquAct can solve complex tasks with $SE(3)$ variation. Additional experiments empirically validate that EquAct is robust to distractors, resilient to occlusion, and exhibits low equivariance error.

There are several limitations of EquAct. Firstly, the keyframe action formulation assumes the task can be solved by several key gripper poses. This assumption is satisfied in RLBench tasks but could be broken in fine-graind manipulation settings (Chi et al., 2023). Moreover, despite EquAct scales well with training data in Table 1, the data efficiency and semantic generalization could be further improved by leveraging pre-trained vision models (Radford et al., 2021; Shafiullah et al., 2022; Gervet et al., 2023). Lastly, the training and the inference speed of EquAct is slower than the best baseline; a more efficient equivariant backbone can speed up the inference.

ACKNOWLEDGMENTS

We would like to thank Hyunwoo Ryu, David Klee, as well as all members of the Helping Hands Lab for their valuable discussions. This work was supported in part by NSF grants 2107256, 2134178, 2314182, 2409351, 2442658, and NASA grant 80NSSC19K1474.

REPRODUCIBILITY STATEMENT

Our implementation is available in `https://github.com/ZXP-S-works/EquAct`. Hyperparameters for all simulation experiments are detailed in Appendix G.

ETHICS STATEMENT

This research employs only publicly available datasets released under appropriate licenses with publisher ethical approval. We collect no personally identifiable information and use no harmful or sensitive data. All work is conducted for academic research purposes.

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

## A   THE USE OF LLM

In this work, we utilize large language models (LLMs) exclusively for language polishing and refinement of our written content. We do NOT employ LLMs for any other aspects of this research, including but not limited to: conceptual development, experimental design, data analysis, result interpretation, literature review, or generation of core research content. All substantive intellectual contributions, methodological innovations, and scientific insights are entirely our own work.

## B   PROOFS

### B.1   PROOF OF PROPOSITION 4.1:

*Proof.* To prove the equivariance of EquAct with respect to $o$, we only need to prove that every layer of EquAct is equivariant, then by induction, EquAct is equivariant to the observation $o$. See Proof B.2, B.3 that proves the equivariance of the proposed spherical maxpool layers and the proposed spherical upsampling layers. Referring (Liao et al., 2024) for proof of Equiformer layers and (Cohen et al., 2018) for proof of Spherical CNNs.

To prove the invariance of EquAct with respect to the nature language instruction $n$, we only need to prove that the iFiLM layers are invariance to $n$, see Proof B.4. $\square$

### B.2   PROOF OF PROPOSITION 4.2:

*Proof.* Focusing on the right-hand side of Equation 4, and denoting the point with the largest manganite of Fourier coefficients after transformation $g = r \ltimes t$ as $p_g^*$:

$$\text{smaxpool}\{\text{D}(r) \cdot c_{l,p} | p \in t + knn(x)\} = \text{D}(r) \cdot c_{l,p_g^*} \tag{11}$$

Expanding the equation of $p_g^*$, and using the property that the winger-D matrices are orthogonal, we have:

$$p_g^* = \arg\max_{p \in t+knn(x)} \|\text{D}(r) \cdot c_{l,p}\|_2^2 \tag{12}$$
$$= \arg\max_{p \in t+knn(x)} \left( (\text{D}(r) \cdot c_{l,p})^\text{T} (\text{D}(r) \cdot c_{l,p}) \right) \tag{13}$$
$$= \arg\max_{p \in t+knn(x)} (c_{l,p}^\text{T} \cdot \text{D}(r)^\text{T} \text{D}(r) \cdot c_{l,p}) \tag{14}$$
$$= \arg\max_{p \in t+knn(x)} (c_{l,p}^\text{T} c_{l,p}) \tag{15}$$
$$= \arg\max_{p \in t+knn(x)} \|c_{l,p}\|_2^2 \tag{16}$$
$$= t + \arg\max_{p \in knn(x)} \|c_{l,p}\|_2^2 \tag{17}$$
$$= t + p^* \tag{18}$$

Thus:

$$\text{smaxpool}\{\text{D}(r) \cdot c_{l,p} | p \in t + knn(x)\} = \text{D}(r) \cdot c_{l,p_g^*} = \text{D}(r) \cdot c_{l,t+p^*} = \text{D}(r) \cdot c'_{l,t+x} \tag{19}$$

$\square$

### B.3 PROOF OF PROPOSITION 4.3:

*Proof.* Expanding the right-hand side of Equation 6 gives:

$$\sup\{D(r) \cdot c_{l,p'}, t + x | p' \in t + \text{knn}(x)\} = \text{softmax}_{p' \in t + \text{knn}(x)} \left(\frac{1}{\|t + x - p'\|}\right) D(r) \cdot c_{l,p'} \quad (20)$$

$$= \text{softmax}_{t+p \in t + \text{knn}(x)} \left(\frac{1}{\|t + x - t - p\|}\right) D(r) \cdot c_{l,t+p} \quad (21)$$

$$= \text{softmax}_{p \in \text{knn}(x)} \left(\frac{1}{\|x - p\|}\right) D(r) \cdot c_{l,t+p} \quad (22)$$

$$= D(r) \cdot \text{softmax}_{p \in \text{knn}(x)} \left(\frac{1}{\|x - p\|}\right) c_{l,t+p} \quad (23)$$

$$= D(r) \cdot c'_{l,t+x} \quad (24)$$

Line 23 is due to Schur's lemma (Schur, 1905), which proved that any linear combination of Fourier coefficients is equivariant. ☐

### B.4 PROOF OF PROPOSITION 4.4:

*Proof.* When $l = 0$, the wingle-D matrix is an identity matrix, thus:

$$\beta\big(D(r) \cdot c_0\big) + \gamma = \beta c_0 + \gamma = c'_0 = D(r) \cdot c'_0 \quad (25)$$

When $l > 0$, expanding the right-hand side of Equation 8 and applying Schur's lemma (Schur, 1905) we have:

$$\alpha_l\big(D(r) \cdot c_l\big) = D(r) \cdot (\alpha_l c_l) = D(r) \cdot c'_l \quad (26)$$

☐

## C ADDITIONAL BACKGROUND ON KEYFRAME IMITATION LEARNING AND MULTI-TASK MANIPULATION POLICY.

The keyframe action formulation (James & Davison, 2022; James et al., 2022) defines the setting where the policy predicts the next goal pose of the gripper based on the current observation. A motion planner then generates a collision-free trajectory to reach this predicted goal. This formulation decomposes complex trajectories into a sequence of keyframe poses, thereby simplifying policy learning while preserving the ability to solve a wide range of manipulation tasks. Building on this, keyframe imitation learning (Shridhar et al., 2023) formulates the problem as imitation learning, where the policy $\pi(o) = a$ learns to predict the expert keyframe action $a$ given an observation $o$ from expert demonstrations. Multi-task keyframe manipulation policies (Shridhar et al., 2023; Goyal et al., 2023; Gervet et al., 2023; Goyal et al., 2024; Ke et al.) extend this formulation to support multiple skills by conditioning the policy on natural language goals $n$, enabling task-specific behavior across a diverse set of instructions.

## D 18 RLBENCH TASKS WITH STANDARD AND $\text{SE}(3)$ INITIALIZATIONS

The 18 RLBench tasks Shridhar et al. (2023); James et al. (2020) are initialized with objects in random $\text{SE}(2)$ poses. In this paper, we present 18 RLBench tasks with $\text{SE}(3)$ variation, where in addition to the $\text{SE}(2)$ initialization, the pose of objects are further perturbed with $\text{SO}(3)$ transformation. This change will leads keyframe actions change in $\text{SE}(3)$. For detailed $\text{SO}(3)$ perturbation range and perturbed object, see Table 6.

## E EQUIVARIANCE ERROR

Table 6: **18 Language-conditioned tasks in RLBench** (James et al., 2020) with SE(3) initializations.

| Task | Variation Type | Perturbed Object | SO(3) Perturbation $(r, p)$ | Language Template |
|------|----------------|------------------|------------------------------|-------------------|
| open drawer | placement | drawer | $[0, -0.5], [0.6, 0.5]$ | "open the __ drawer" |
| slide block | color | plane | $[-0.12, -0.12], [0.12, 0.12]$ | "slide the block to __ target" |
| sweep to dustpan | size | broom holder | $[0, 0, -0.9], [0, 0, 0.9]$ | "sweep dirt to the __ dustpan" |
| meat off grill | category | grill table | $[-0.25, -0.25], [0.25, 0.25]$ | "take the __ off the grill" |
| turn tap | placement | tap | $[-0.5, -0.5], [0.5, 0.5]$ | "turn __ tap" |
| put in drawer | placement | drawer | $[0, -0.2], [0.2, 0.2]$ | "put the item in the __ drawer" |
| close jar | color | jar | $[0, -0.5], [0.6, 0.5]$ | "close the __ jar" |
| drag stick | color | plane | $[-0.12, -0.12], [0.12, 0.12]$ | "use the stick to drag the cube onto the __ target" |
| stack blocks | color, count | plane | $[-0.15, -0.15], [0.15, 0.15]$ | "stack __ __ blocks" |
| screw bulb | color | lamp base | $[-0.6, -0.6], [0.6, 0.6]$ | "screw in the __ light bulb" |
| put in safe | placement | safe | $[-0.25, -0.3], [0.5, 0.3]$ | "put the money away in the safe on the __ shelf" |
| place wine | placement | wine rack | $[-0.5, -0.5], [0.5, 0.5]$ | "stack the wine bottle to the __ of the rack" |
| put in cupboard | category | cupboard | $[-0.5, -0.5], [0.5, 0.5]$ | "put the __ in the cupboard" |
| sort shape | shape | shape sorter | $[-0.25, -0.25], [0.25, 0.25]$ | "put the __ in the shape sorter" |
| push buttons | color | buttons | $[-0.25, -0.25], [0.25, 0.25]$ | "push the __ button, [then the __ button]" |
| insert peg | color | pillars | $[-0.3, -0.4], [0.3, 0.4]$ | "put the ring on the __ spoke" |
| stack cups | color | cups | $[-0.3, -0.3], [0.3, 0.3]$ | "stack the other cups on top of the __ cup" |
| place cups | count | cups | $[0, -0.5], [0.6, 0.5]$ | "place __ cups on the cup holder" |

We empirically measure the degree of equivariance error in our model. This is defined as the geodesic distance $d$ based on Equation 2:

$$equ\_error = d(\pi(g \cdot o, n), g \cdot \pi(o, n))$$

Here, $g$ rotates the point cloud and proprioceptive information in the observation and the predicted action. $g$ is uniformly sampled from a translational range of $0.1$ m along the X, Y, and Z axis, and a rotational range of $60°$ along the roll and pitch axis, and full $360°$ around the yaw axis. We compared the equivariance error in EquAct with that in 3DDA (Ke et al.). Both models are those used to produce the results in Table 1, and both are trained with 100 demonstrations. We focus on the "open_drawer" task because its action is uni-modal, which minimizes ambiguity from multi-modal action spaces and allows for a clean measurement of equivariance error. Our results in Table 7 indicate that our model has lower equivariance error than 3DDA, empirically verifying EquAct is equivariant.

Table 7: Empirical equivariance error, reported as geodesic distances (in meters for translation and radians for rotation). Identity setting measures the randomness of inference, and SE(3) perturbation measures equivariance error.

| Model | Identity | SE(3) Perturbation |
|-------|----------|--------------------|
| EquAct | 0.09 rad, 0.004 m | 0.8 rad, 0.038 m |
| 3DDA | 0.07 rad, 0.006 m | 1.4 rad, 0.132 m |

# F    DETAILS OF 4 PHYSICAL TASKS

Our real-world experiments are carried out on a UR5 robotic arm equipped with a Robotiq 2F-85 gripper and three Intel RealSense D455 cameras (front, left, and right cameras), as shown in Figure 6. Keyframe actions are collected using a 6-DoF 3DConnexion SpaceMouse. We collect both visual observations (from all three cameras) and robot end-effector actions (position, orientation, and gripper states). During training and evaluation, we randomize the objects' SE(3) (or SE(2)) poses by physically rotating the object—clamped on the magic-arm camera mount—before each rollout.

The 4 physical tasks are visualized in Figure 5. For details of these tasks, see descriptions below.

## F.1    DISASSEMBLE PIPE

**Task:** Disassemble the required link of the pipe: first, second, all.

**Number of keyframe actions:** 3-10.

**Variations:** "first link", "second link","all".

**Objects:** Pipes consisting of five sections of water pipes.

**Success Metric**: The robot must accurately grasp the target pipe segment and completely remove it from the intact assembly.

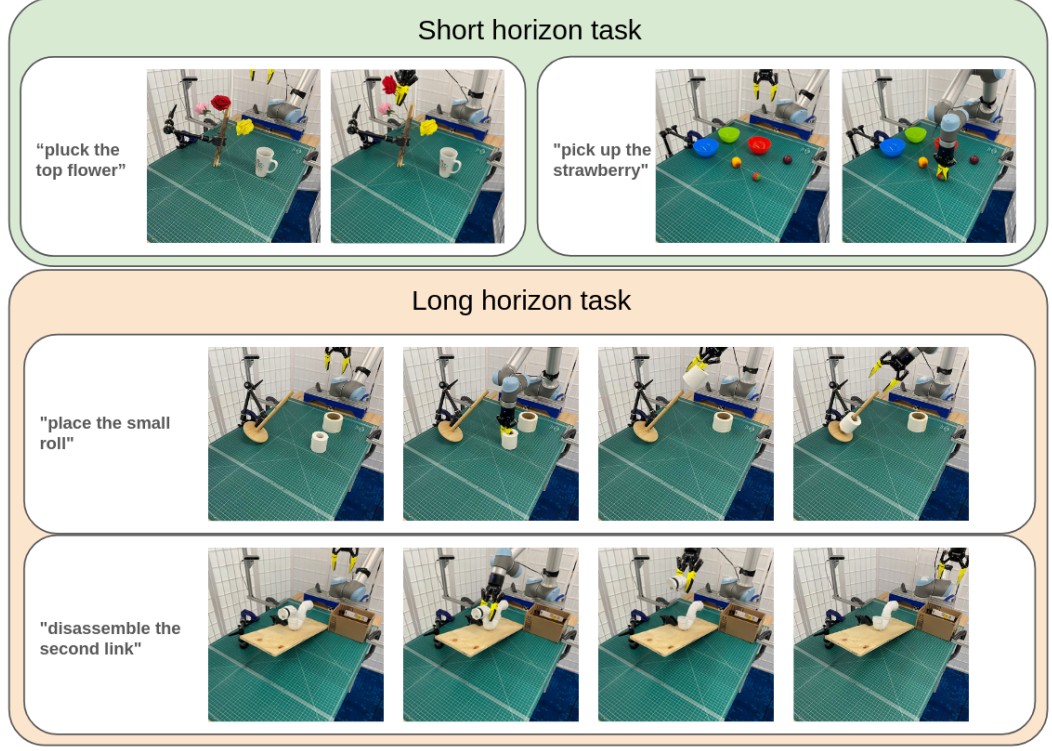

Figure 5: 4 **Physical tasks.**

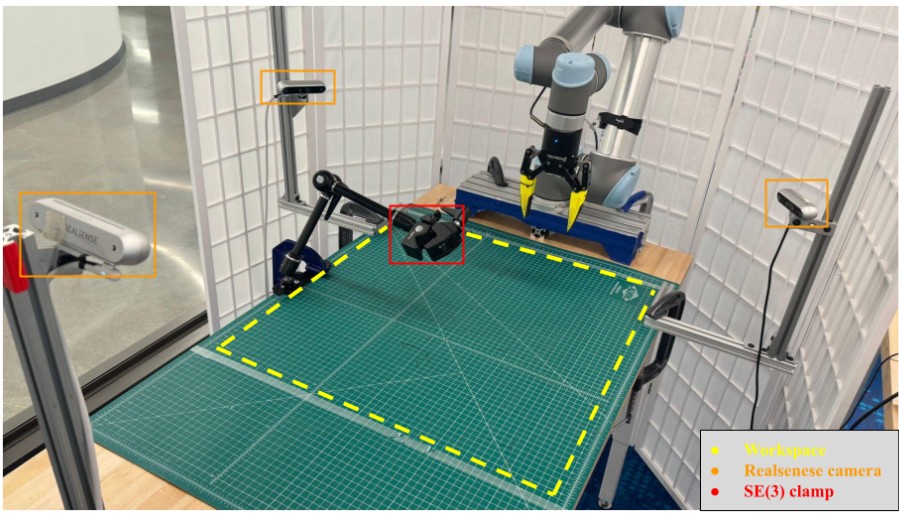

Figure 6: **Real world experimental setup**

## F.2 PLUCK FLOWERS

**Task:** Pluck the specified flower: top, middle, bottom.

**Number of keyframe actions:** 4.

**Variations:** "top flower", "middle flower", "bottom flower".

**Objects:** Three artificial flowers and one vase.

**Success Metric:** The robot must accurately grab the designated flower and pluck it.

### F.3 PICK FRUIT

**Task:** Pick up the specified fruit(strawberry,peach,plum).

**Number of keyframe actions:** 3.

**Variations:** "strawberry", "peach", "plum".

**Objects:** Three fruits of mixed types.

**Success Metric:** The robot must correctly identify, grasp the target fruit.

### F.4 INSTALL TOILET ROLL

**Task:** Place the specified toilet paper roll: large, small.

**Number of keyframe actions:** 5.

**Variations:** "large roll", "small roll".

**Objects:** Two toilet-paper rolls and one wall-mounted holder.

**Success Metric:** The robot must pick up the specified roll and mount it onto the holder.

## G HYPERPARAMETERS

We report the following hyperparameters in Table 8 for EquAct as well as baselines we compared in the paper. The differences in training iterations across baselines are primarily due to variations in computational resources (number of GPUs), which in turn affect batch sizes. For instance, EquAct was trained on a single GPU with a batch size of 2, so it is trained with 8e5 iterations, whereas SAM2Act was trained on 32 GPUs with a batch size of 256, so its iteration is reduced to 5e4.

Table 8: **Hyperparameters.** sim: the hyperparameters used in simulation experiments. phy: the hyperparameters used in physical experiments.

| Name of the hyperparameter | EquAct (sim) | EquAct (phy) | Method 3DDA (sim) | 3DDA (phy) | SAM2ACT (sim) |
|---|---|---|---|---|---|
| # $a_t$ (train/test) | 450/3000 | 450/6000 | None | None | None |
| $a_t$ coarse2fine levels | 3 | 3 | None | None | None |
| # $a_r$ (train/test) | 36,864/2,359,296 | 36,864/2,359,296 | None | None | None |
| learning rate | 1e-4 | 1e-4 | 1e-4 | 1e-4 | 1e-4 |
| lr scheduler | None | None | None | None | cosine |
| batch size | 2 | 2 | 7 | 7 | 8 |
| training iterations | 8e5 | 6e4 | 6e5 | 1.2e5 | 5.625e4 |

