# OpenReview forum: "EquAct: An SE(3)-Equivariant Multi-Task Transformer for 3D Robotic Manipulation"
_ICLR.cc/2026/Conference — ICLR 2026 Poster_

### Official Review · Reviewer_UBsD · 2025-10-26

**Soundness:** 3
**Presentation:** 3
**Contribution:** 3
**Rating:** 6
**Confidence:** 5

**Summary:**

This paper proposes a multi-task SE(3) equivaraint robot policies.  The model consists of (1) a SE(3)-equivariant encoder that transforms input point cloud into spherical Fourier features, followed by (2) SE(3)-invariant feature-wise linear modulation layer that facilitates conditioning on task instructions.  The proposed method sets a new state-of-the-art on RLBench simulation benchmark, while demonstrating strong generalization to spatial variety.

**Strengths:**

1. The proposed method sets a new state-of-the-art on the challenging RLBench simulation benchmark.  In particular, it demonstrates superior performance on testing scenes where objects are randomly SE(3) initialized.
2. The design choice of using SE(3)-equivariant point cloud encoder is techinically sound.  Its efficacy is also supported by the ablation study.
3. This paper reports both training time, inference speed and required GPU mem, facilitating better understanding of the proposed method.
4. The authors have released the code base, facilitating reproducibility of the proposed method.

**Weaknesses:**

1. This paper only evaluates the proposed method on one RLBench simulation benchmark and a real-world benchmark.  I'd highly encourage the authors to evaluate on other benchmarks, such as SimplerEnv [1] or LIBERO [2] or CALVIN [3], to further demonstrate the generality and effectiveness of the proposed method .
2. It's widely known that 3DDA has poor sim2real generalization due to the visual appearance gap between the simulation and the real world.  This paper uses a SE(3)-equivariant point cloud encoder, and point cloud has stronger sim2real generalization than RGB image encoders.  Along with the SE(3)-equivariance, it'd be interesting to see if the proposed method trained in simulation can be directly deployed in the real world.  Such results would intrigue the community and attract more attention in SE(3)-equivariant policies.

---

Reference

[1] Li, Xuanlin, et al. "Evaluating real-world robot manipulation policies in simulation." arXiv preprint arXiv:2405.05941 (2024).

[2] Liu, Bo, et al. "Libero: Benchmarking knowledge transfer for lifelong robot learning." Advances in Neural Information Processing Systems 36 (2023): 44776-44791.

[3] Mees, Oier, et al. "Calvin: A benchmark for language-conditioned policy learning for long-horizon robot manipulation tasks." IEEE Robotics and Automation Letters 7.3 (2022): 7327-7334.

**Questions:**

1. What is the performance of EquAct on SimplerEnv or LIBERO or CALVIN simulation benchmark?
2. How does the proposed method work in sim2real setup?

I'm open to adjust my rating based on the author's rebuttal.

---

> ### Author Response · Authors · 2025-12-01
> **Response**
>
> We thank the reviewer for the careful review. Our responses are provided below.
> > What is the performance of EquAct on SimplerEnv or LIBERO or CALVIN simulation benchmark?
>
> EquAct is built on a keyframe-action formulation, which is not directly compatible with SimplerEnv, LIBERO, or CALVIN. As noted in Section 6, EquAct can be integrated with diffusion policies in a hierarchical framework to handle tasks in these simulation environments.
>
> > How does the proposed method work in sim2real setup?
>
> Due to the hardware mismatch—18RLBench uses a Panda manipulator while our real-world experiments use a UR5—we did not conduct sim-to-real experiments. Nevertheless, our physical experiments (Table 5) show that EquAct remains robust in the presence of distractor objects, indicating that it can effectively focus on task-relevant features while ignoring irrelevant distractions. This could benefit sim2real transfer.

---

### Official Review · Reviewer_xNKT · 2025-10-28

**Soundness:** 3
**Presentation:** 3
**Contribution:** 3
**Rating:** 8
**Confidence:** 4

**Summary:**

The paper proposes a SE(3)-equivariant multi-task policy network with several architectural designs: (i) an equivariant U-net, (ii) an invariant FiLM layer, and (iii) an equivariant field network. The equivariance is mathematically proven. The network shows good performance in both simulation and real-world experiments.

**Strengths:**

- Multi-task learning with generalization is a much harder problem than per-task learning, while most prior works with equivariance could only tackle the latter.
- The proposed network has both theoretical foundations to guarantee its equivariance and engineering designs to improve its performance and efficiency.
- The experiments are well-designed and well-conducted. Especially some small but important details are well-studied in the experiments:
	- I appreciate that the authors also show the training/inference time and memory consumption for the proposed network, and it's good that the results match the baselines -- usually, equivariant networks can easily consume more time or memory as a cost for their "for-free" generalization to unseen poses
	- The robustness and empirical equivariance errors are tested.

**Weaknesses:**

- Some details, especially the experimental setups, can be made clearer. See the minor points and questions below.


Some minor points:
- Line 157: $x = \{x_T, x_{\text{open}\}, x=\{e, a\}$ looks confusing at first glance -- I understood it after a while, but I think it can be expressed in a clearer way.
- Table 1 is a bit hard to read at first, and it took me a while to understand its configuration: (i) The two small regions for the avg. success rate and time/memory consumptions, perhaps mark them in different background colors, so that they look more distinct from the per-task success rates. (ii) "10* 10 100" looks too abbreviated for training setups, maybe make it a bit more concrete, like "SE(2)/10" or something, and I think the table still has some space for that.
- Perhaps say "real-world experiments" instead of "physical tasks"? It made me confused before reading Appendix F -- what are "physical tasks"? force control? physical parameter related? ...

**Questions:**

- How are the query actions sampled when training the equivariant field network? Is it a uniform sampling? Is the sampling space too large: rotation + translation, in a large scene?
- In Section 4, it says: "During training, we also augment the dataset with respect to equation 1 by randomly rotating the
point cloud and the action simultaneously with [±5◦,±5◦,±45◦] rotation along [x, y, z] axis." How does it relate to the SE(3)/SE(2) randomizations in Section 5.1 Experiment settings (line 409-414)? Also, why are these numbers chosen?

---

> ### Author Response · Authors · 2025-12-01
> **Response**
>
> We thank the reviewer for their careful review. Our responses are provided below.
>
> > Some details, especially the experimental setups, can be made clearer. See the minor points and questions below.
>
> We appreciate the reviewer’s comments and have improved the presentation in the updated draft.
> Questions:
>
> > How are the query actions sampled when training the equivariant field network? Is it a uniform sampling? Is the sampling space too large: rotation + translation, in a large scene?
>
> The query actions are uniformly sampled. Specifically, a_t contains 449 uniformly sampled translational actions along with ground-truth actions (150 actions across 3 coarse-to-fine levels); ara_rar​ contains 36,864 rotational actions sampled using the HEALPix level-3 grid; and a_open​ contains 2 gripper actions (open and close). The sampling space is identical across all tasks. We have added this clarification to Section 4.
>
> > In Section 4, it says: "During training, we also augment the dataset with respect to equation 1 by randomly rotating the point cloud and the action simultaneously with [±5◦,±5◦,±45◦] rotation along [x, y, z] axis." How does it relate to the SE(3)/SE(2) randomizations in Section 5.1 Experiment settings (line 409-414)? Also, why are these numbers chosen?
>
> We would like to clarify that the data augmentation is applied directly to the dataset. The SE(3)/SE(2) random initialization refers to the initial object poses in the simulator. The data-augmentation ranges follow the settings used in SAM2ACT. We have added this clarification to Table 1.

---

### Official Review · Reviewer_Nxid · 2025-10-30

**Soundness:** 2
**Presentation:** 3
**Contribution:** 2
**Rating:** 4
**Confidence:** 4

**Summary:**

This paper proposes EquAct, an SE(3)-equivariant multi-task Transformer for open-loop robotic manipulation.
The method introduces explicit geometric inductive bias into 3D point cloud and language-conditioned policy learning, aiming to improve generalization and sample efficiency under spatial perturbations.
The model consists of three main components: (1) an SE(3)-equivariant Point Transformer U-Net for geometric encoding, (2) an Invariant Feature-wise Linear Modulation (iFiLM) layer for language conditioning, and (3) an Equivariant Field Network for action candidate evaluation.
Experiments on 18 RLBench tasks and several real-robot setups demonstrate clear advantages over non-equivariant baselines under SE(3)/SE(2) perturbations. The paper also provides quantitative analyses of equivariance error.

**Strengths:**

(1)The work systematically integrates explicit SE(3) equivariance into a multi-task vision-language manipulation Transformer with modular and theoretically grounded design.
(2)The introduction of SE(3) initialization in RLBench contributes an independent and valuable benchmark for studying spatial generalization.
(3)The method achieves strong performance under SE(3)/SE(2) perturbations, demonstrating the benefits of geometric inductive bias in low-data regimes.

**Weaknesses:**

(1)Lack of comparison with other equivariant models. The paper only compares with non-equivariant baselines, leaving unclear whether EquAct’s design offers advantages within the broader class of equivariant inductive biases.
(2)It is unclear whether explicit SE(3) constraints remain beneficial when data scale or pretraining increases, or if the performance gains would saturate.
(3)In real robotic systems, camera pose variations, viewpoint shifts, and lens distortions break strict SE(3) symmetry. The model’s robustness to such non-ideal conditions has not been examined.

**Questions:**

1. Could the authors include comparisons or analyses with other SE(3)-equivariant models such as [1][2]? This is not mandatory, but discussing why previous methods are limited to single-task settings while EquAct handles multi-task scenarios would clarify whether this advantage arises from the keyframe-based formulation rather than the equivariance design itself.
2. When scaling to larger datasets, does the model still rely on explicit equivariance for good performance? Are there signs of saturation?
3. In real-world conditions, camera viewpoint shifts and optical distortions are common. It would be valuable to include experiments testing robustness under such non-ideal equivariance scenarios.
If the authors can effectively address these concerns, I would consider raising the score.
References
[1] Tie, Chenrui, et al. “Et-seed: Efficient trajectory-level SE(3) equivariant diffusion policy.” arXiv preprint arXiv:2411.03990 (2024).
[2] Qi, Yu, et al. “Two by two: Learning multi-task pairwise objects assembly for generalizable robot manipulation.” CVPR 2025.

---

> ### Author Response · Authors · 2025-12-02
> **Response**
>
> We thank the reviewer for the careful review. Our responses are provided below.
>
> > Could the authors include comparisons or analyses with other SE(3)-equivariant models such as [1][2]? This is not mandatory, but discussing why previous methods are limited to single-task settings while EquAct handles multi-task scenarios would clarify whether this advantage arises from the keyframe-based formulation rather than the equivariance design itself.
>
> To evaluate the advantages of EquAct over prior equivariant architectures, we implemented VN-DGCNN [1,2] on RLBench tasks as a baseline. We do not include SE(3)-Transformer due to its prohibitively high computational cost stemming from full tensor-product operations [3,4]. In the vector-neuron baseline, we replace our SE(3)-equivariant Point Transformer U-Net with a VN-DGCNN model. The results, reported in Table R1, show that although VN-DGCNN is SE(3)-equivariant, it underperforms EquAct by more than 30%. A likely reason is that VN-DGCNN operates only on vector features (type-1 irreps), whereas EquAct leverages higher-order features (up to type-3). These higher-order components allow EquAct to represent signals at finer angular resolutions, leading to significantly improved performance.
>
> | Method                 | avg. SR ↑ | place wine | place cups | reach drag | insert peg |
> |-----------------------------|-----------|------------|------------|------------|-------------|
> | **Ours**                        | **52.8**  | 45         | 62         | 90         | 14          |
> | VN-DGCNN	    | 22	        |   64       | 5          |  5          | 14         |
>
> [1] Qi, Yu, et al. “Two by two: Learning multi-task pairwise objects assembly for generalizable robot manipulation.” CVPR 2025.
>
> [2] Congyue Deng, et al. “Vector Neurons: A General Framework for SO(3)-Equivariant Networks.” ICCV 2021.
>
> [3] Saro Passaro, et al. Reducing SO(3) Convolutions to SO(2) for Efficient Equivariant GNNs. ICML, 2023.
>
> [4] Tie, Chenrui, et al. “Et-seed: Efficient trajectory-level SE(3) equivariant diffusion policy.” arXiv preprint arXiv:2411.03990 (2024).
>
> > When scaling to larger datasets, does the model still rely on explicit equivariance for good performance? Are there signs of saturation?
>
> We conduct a data-scaling experiment on the place_cups task. We observe that EquAct’s performance improves as the amount of training data increases from 10 to 100 demonstrations, but saturates when further scaling from 100 to 1000. We hypothesize that the bottleneck arises from the keyframe action formulation: the coarse keyframe control can cause unintended collisions between the cup and the mug tree, limiting the benefits of additional data. Nevertheless, prior work has shown that when data scales, equivariant models consistently outperform their non-equivariant counterparts [1].
> | # demos | 10 | 100 | 1000 |
> |------------|------|-----|-----|
> | EquAct  | 58  | 71 | 71 |
>
> [1] Johann Brehmer, et al. Does equivariance matter at scale? Arxiv 2024.
>
> > In real-world conditions, camera viewpoint shifts and optical distortions are common. It would be valuable to include experiments testing robustness under such non-ideal equivariance scenarios. If the authors can effectively address these concerns, I would consider raising the score. References [1] Tie, Chenrui, et al. “Et-seed: Efficient trajectory-level SE(3) equivariant diffusion policy.” arXiv preprint arXiv:2411.03990 (2024). [2] Qi, Yu, et al. “Two by two: Learning multi-task pairwise objects assembly for generalizable robot manipulation.” CVPR 2025.
>
> Due to limited time and resources, we were unable to evaluate robustness under camera viewpoint shifts or lens distortion. However, Table 6 includes an ablation that mimics real-world asymmetries: all baselines are trained using point clouds reconstructed from a single camera, resulting in significant occlusions. Despite these imperfect and symmetry-breaking observations, EquAct exhibits only a modest performance drop of 5.8%.

---

### Official Review · Reviewer_cZ5N · 2025-11-01

**Soundness:** 4
**Presentation:** 4
**Contribution:** 4
**Rating:** 10
**Confidence:** 3

**Summary:**

The paper proposes a novel approach for adapting visuomotor policies to be SE(3) equivariant to scene changes and SE(3) invaraiant to similar language commands. This is achieved by using an SE(3)-equivariant point transformer that adapts to changes in the observation space, while having an SE(3)-invariant FiLM layers to adapt (or consequently not adapt) to changes in the language commands. They authors show the improved performance of the proposed approach for learning a next-best keyframe manipulation policy in a multi-task imitation learning setting.

**Strengths:**

The proposed approach does a good job at ensuring SE(3) equivariance/invariance through different parts of the system, such as the equivariant transformer arhitechture for encoding the points, and the Field network. The proposed novelties are experimentally evaluated and show good performance especially under high SE(3) variability with just a low number of samples (10 demos) and scales well with more number of demonstrations.

**Weaknesses:**

The policy architecture and the corresponding action refinement/selection is unclear. How is the implicit action value function used to predict the actions? Is it by randomly sampling actions and evaluating them? Or is it by having a separate policy network trained in an actor-critic approach?
Training details about the implementation of the cross entropy losses are missing to make the paper more complete and to improve the reproducibility aspects of the paper.
Some discussion on not using direct behavioural cloning instead of an implicit action-value function would benefit the paper.

**Questions:**

In Lines 201-202, does the Q function take in a single action sample? Why does the gripper Q function take in only the translational action and not the rotational one as well? Moreover, how are the cross-entropy losses calculated?
The SO(3) Perturbation (r, p) is unclear in Table 4. are they the deviations along XYZ and roll pitch yaw? if so, should it not be SE(3) Perturbations in that case?

---

> ### Author Response · Authors · 2025-12-01
> **Response**
>
> We thank the reviewer for their careful review. Our responses are provided below.
>
> > In Lines 201-202, does the Q function take in a single action sample?
>
> We would like to clarify that the Q-function is evaluated over a large set of uniformly sampled actions. Specifically, a_t​ includes 449 uniformly sampled translational actions together with the ground-truth actions (150 actions × 3 coarse-to-fine levels); a_r​ includes 36,864 rotational actions sampled using the HEALPix level-3 grid; and a_open​ includes 2 actions (open and close). We have added this clarification to Section 4 in the revised draft.
>
> > Why does the gripper Q function take in only the translational action and not the rotational one as well?
>
> While EquAct can process rotational actions in the gripper Q-function, we found empirically that translational actions alone are sufficient.
>
> > Moreover, how are the cross-entropy losses calculated?
>
> When computing the cross-entropy loss, the action closest to the ground-truth action is assigned as the positive label, and all remaining actions are treated as negative samples.
>
> > The SO(3) Perturbation (r, p) is unclear in Table 4. are they the deviations along XYZ and roll pitch yaw? if so, should it not be SE(3) Perturbations in that case?
>
> The perturbations (r, p) represent deviations along the roll and pitch axes. For yaw, we follow the original perturbation settings provided in 18RLBench for each task to maintain consistency.

---

### Author Response · Authors · 2025-12-03
**Summary of response II:**

## For reviewer UBsD (rating 6):

**What is the performance of EquAct on SimplerEnv or LIBERO or CALVIN simulation benchmark?**

EquAct is built on a keyframe-action formulation, which is not directly compatible with SimplerEnv, LIBERO, or CALVIN. As noted in Section 6, EquAct can be integrated with diffusion policies in a hierarchical framework to handle tasks in these simulation environments.

**How does the proposed method work in sim2real setup?**

Due to the hardware mismatch—18RLBench uses a Panda manipulator while our real-world experiments use a UR5—we did not conduct sim-to-real experiments. Nevertheless, our physical experiments (Table 5) show that EquAct remains robust in the presence of distractor objects, indicating that it can effectively focus on task-relevant features while ignoring irrelevant distractions. This could benefit sim2real transfer.

---

### Author Response · Authors · 2025-12-03
**Summary of response I**

## For reviewer cZ5N (rating 10):

**What is the input to the Q function?**

The input consists of uniformly sampled actions on the translational and rotational actions.  For the gripper Q function, we empirically find that not taking in rotational information is sufficient. We have added the detailed classification to Section 4 in the revised draft.

**What is the calculation of cross-entropy loss?**

For the loss calculation, the action closest to the ground-truth action is assigned as the positive label, while other actions are treated as negative samples.

**What is the definition of the SO(3) perturbation of the objects?**

The perturbations (r, p) represent deviations along the roll and pitch axes. For yaw, we follow the original perturbation settings provided in 18RLBench for each task to maintain consistency.

## For reviewer Nxid (rating 4):

**Could the authors include comparisons with other SE(3)-equivariant models?**

We implemented VN-DGCNN [1,2] on RLBench tasks as a baseline. We do not include SE(3)-Transformer due to its prohibitively high computational cost [3,4]. In the vector-neuron baseline, we replace our SE(3)-equivariant Point Transformer U-Net with a VN-DGCNN model.

The results, reported in Table R1, show that although VN-DGCNN is SE(3)-equivariant, it underperforms EquAct by more than 30%. A likely reason is that VN-DGCNN operates only on vector features (type-1 irreps), whereas EquAct leverages higher-order features (up to type-3). These higher-order components allow EquAct to represent signals at finer angular resolutions, leading to significantly improved performance.

| Method                 | avg. SR ↑ | place wine | place cups | reach drag | insert peg |
|--------------------------------|-----------|------------|------------|------------|-------------|
| **Ours**                        | **52.8**  | 45         | 62         | 90         | 14          |
| Vector Neuron	   | 22	        |   64       | 5          |  5          | 14         |

[1] Qi, Yu, et al. “Two by two: Learning multi-task pairwise objects assembly for generalizable robot manipulation.” CVPR 2025.

[2] Congyue Deng, et al. “Vector Neurons: A General Framework for SO(3)-Equivariant Networks.” ICCV 2021.

[3] Saro Passaro, et al. Reducing SO(3) Convolutions to SO(2) for Efficient Equivariant GNNs. ICML, 2023.

[4] Tie, Chenrui, et al. “Et-seed: Efficient trajectory-level SE(3) equivariant diffusion policy.” arXiv preprint arXiv:2411.03990 (2024).

**Does the model scale to larger datasets?**

We conduct a data-scaling experiment on the place_cups task. We observe that EquAct’s performance improves as the amount of training data increases from 10 to 100 demonstrations, but saturates when further scaling from 100 to 1000. We hypothesize that the bottleneck arises from the keyframe action formulation. Nevertheless, prior work has shown that when data scales, equivariant models consistently outperform their non-equivariant counterparts [1].

| # demos | 10 | 100 | 1000 |
|------------|------|-----|-----|
| EquAct   | 58  | 71 | 71 |

[1] Johann Brehmer, et al. Does equivariance matter at scale? Arxiv 2024.

**It would be valuable to include experiments testing robustness under non-ideal equivariance scenarios.**

Due to limited time and resources, we were unable to evaluate robustness under camera viewpoint shifts or lens distortion. However, Table 6 includes an ablation that mimics real-world asymmetries: all baselines are trained using point clouds reconstructed from a single camera, resulting in significant occlusions. Despite these imperfect and symmetry-breaking observations, EquAct exhibits only a modest performance drop of 5.8%.

## For reviewer xNKT (rating 8):
**Some details, especially the experimental setups, can be made clearer.**

We appreciate the reviewer’s comments and have improved the presentation in the updated draft. Specifically, we updated:
- End-effector station/action definition on line 157-159.
- Table1 with highlight color and experiment description.

**How are the query actions sampled when training the equivariant field network?**

The query translational and rotational actions are uniformly sampled. The sampling space is identical across all tasks. We have added this clarification to Section 4.

**Confusion of the SE(3)/SE(2) randomizations in Section 4 and in Section 5.1 Experiment settings.**

We would like to clarify that in Section 4, the data augmentation is applied directly to the dataset. In Section 5.1, the SE(3)/SE(2) random initialization refers to the initial object poses in the simulator. We have added this clarification to Table 1.

---

### Meta-Review · Area_Chair_c9Yz · 2025-12-30

**Summary:**

The paper proposes an SE(3)-equivariant multi-task transformer for robotic manipulation.
The initial reviewer scores were 10, 4, 8, 6. Reviewers generally acknowledged the novelty of the proposed method and good performance, especially in low-data settings. The main concerns raised included: unclear policy architecture; the lack of comparisons with other equivariant models; whether explicit SE(3) constraints help with more data; and the lack of results on additional simulated benchmarks.

In the rebuttal, the authors provided additional experimental results and clarifications that addressed most of these concerns. While it was not feasible to include comparisons with a broader range of benchmarks, the existing evaluations on RLBench and real-world experiments are solid. Considering the novelty of the method and the strength of the empirical results, the AC recommends accepting the paper.

**Reviewer Concerns:**

See above.

**Reviewer Scores:**

See above.

---

### Decision · Program_Chairs · 2026-01-26

Accept (Poster)